# Maltooligosaccharides: Properties, Production and Applications

**DOI:** 10.3390/molecules28073281

**Published:** 2023-04-06

**Authors:** Mária Bláhová, Vladimír Štefuca, Helena Hronská, Michal Rosenberg

**Affiliations:** Faculty of Chemical and Food Technology, Institute of Biotechnology, Slovak University of Technology, Radlinského 9, 812 37 Bratislava, Slovakia

**Keywords:** maltooligosaccharides, amylases, health benefits, biocatalysis, enzymatic synthesis, downstream processing, immobilization

## Abstract

Maltooligosaccharides (MOS) are homooligosaccharides that consist of 3–10 glucose molecules linked by α-1,4 glycosidic bonds. As they have physiological functions, they are commonly used as ingredients in nutritional products and functional foods. Many researchers have investigated the potential applications of MOS and their derivatives in the pharmaceutical industry. In this review, we summarized the properties and methods of fabricating MOS and their derivatives, including sulfated and non-sulfated alkylMOS. For preparing MOS, different enzymatic strategies have been proposed by various researchers, using α-amylases, maltooligosaccharide-forming amylases, or glycosyltransferases as effective biocatalysts. Many researchers have focused on using immobilized biocatalysts and downstream processes for MOS production. This review also provides an overview of the current challenges and future trends of MOS production.

## 1. Introduction

Maltooligosaccharides (MOS) have unique properties and important industrial applications. According to the International Union of Pure and Applied Chemistry, oligosaccharides are polymers of monosaccharides with a degree of polymerization (DP) between 3 and 10. Therefore, we defined maltooligosaccharides as homooligosaccharides composed of α-glucose monomers linked by α-1,4 glycosidic linkages and with 3–10 glucose monomers (Figure 1). The α-1,4 glycosidic linkage makes maltooligosaccharides digestible in the small intestine by human intestinal glucoamylase-maltase [1] to glucose. However, a study recently showed that MOS undergoes only partial cleavage and serves as a substrate for intestinal microbiota, which is known as a prebiotic effect [2]. Additionally, MOS reduces intestinal pathobiont microorganisms and increases the levels of beneficial bacteria of the genus *Bifidobacterium*, which improves the health of the colon, for example, by improving intestinal peristalsis, and preventing constipation [3]. MOS might even have immunomodulatory effects [4]. Because of these health benefits, MOS can be categorized as functional oligosaccharides. Functional oligosaccharides are non-digestible by human gut enzymes, except for maltooligosaccharides and trehalose [3], and provide health benefits such fiber and prebiotics [5].

MOS have some typical characteristics, such as low osmotic pressure [3], high water-holding capacity [6], mild sweetness, a high moisturizing effect [7], and the ability to inhibit crystallization of sucrose [8]. Thus, they provide several health benefits when they are used in food processing. Maltooligosaccharides also occur naturally in some fermented foods and beverages, such as beer, and they have a positive effect on the stability and quality of beer foam [9]. Besides being used in the food and beverage industry, they are also used in clinical chemistry as substrates for measuring blood glucose levels. Maltotetraose has antibacterial activity against *Erwinia* sp. [10]. Some species of *Erwinia* sp. are plant pathogens that cause plant diseases, such as stalk rot of plants, flowers, and fruits. In that study, the antibacterial properties of glucose, maltose, maltodextrin, and maltotetraose were investigated. Among them, only maltotetraose inhibited *Erwinia* sp. [10], but since the antibacterial properties of other maltooligosaccharides have not been investigated, they might also have undetected inhibitory and antibacterial activities. However, the findings of that study facilitated the application of these oligosaccharides in agriculture.

Despite considerable interest in MOS and their unique and promising properties, no comprehensive review article dealing with the practical aspects of MOS preparation and use has been published recently. There are a very few recent review articles published in this field, two of them dealing only with the preparation of MOS using maltooligosaccharide-forming amylases (MFA-ses) [6,11] and one reporting on the preparation of MOS from cyclodextrins [12]. This paper aimed to fill this gap by providing a review addressing the properties, applications, and preparation of MOS and their practically useful derivatives.

## 2. Properties of Maltooligosaccharides

The physical properties of maltooligosaccharides depend on the number of glucose monomers, which determines, for example, their solubility in water. However, the number of monomers is not the only aspect that affects the properties of the polymers/oligomers. Another aspect is the type of linkage that influences the action of a given molecule, e.g., in the human body [13]. For example, maltooligosaccharides are classified as glycemic carbohydrates since they are broken down in the small intestine due to their breakable linkages (α-1,4). The broken components are absorbed and serve as a source of energy in the form of glucose. On the contrary, non-glycemic carbohydrates (e.g., isomaltooligosaccharides) contain unbreakable bonds (α-1,6), and thus, they cannot be broken down in the upper digestive tract and promote the growth of beneficial bacteria in the colon [3,13]. The free anomeric carbon at the end of the maltooligosaccharide molecule (Figure 1) makes it a reducing saccharide. For characterizing and classifying starch hydrolysis products, including maltooligosaccharides, the dextrose equivalent (DE) is commonly used. The dextrose equivalent is a measure of the reducing power of a given molecule relative to dry weight. DE values range from 0 to 100. Glucose, the end product of the hydrolysis of starch, has a DE value of 100, while starch has a DE value close to 0. The DE values of other starch hydrolysis products range between these values. The DE value is inversely proportional to the degree of polymerization (DP). For example, maltodextrins have a DE value lower than 20 [13,14].

The properties of maltooligosaccharides differ slightly depending on their DP, but in general, they have high solubility, low bulk density, and low osmolality with mild-to-low sweetness [15]. The basic properties and characteristics of individual maltooligosaccharides are shown in Table 1. Although maltononaose and maltodecaose also belong to maltooligosaccharides, they are not found commonly and are used very rarely in industrial applications and, therefore, are not mentioned in Table 1.

## 3. Application of Maltooligosaccharides

Japan is leading country in the production of various oligosaccharides with a strong history in the research of oligosaccharides for foods that started around 1970–1975 [8]. Since then, oligosaccharides have attracted the attention of many researchers. New derivatives of maltooligoasaccharides with unique functional properties and applications are being developed.

The applications of maltooligosaccharides can be divided into two main groups based on their structure. The first group includes maltooligosaccharides in their natural structure, and they are used as a mixture of several maltooligosaccharides or as a single type of molecule. The second group includes derivatives of maltooligosaccharides, which have more specific applications. The potential of these molecules depends on the substituents they possess and also the type of MOS (Figure 2).

### 3.1. Maltooligosaccharides in Their Natural Structure

MOSs are commonly used in the food industry as a “maltooligomer mix”. Their unique properties, such as low sweetness and viscosity, make them an ideal substitute for sucrose and other saccharides [18]. Because of its hygroscopicity, the maltooligosaccharide mix is also used as a moisture regulator of food [19]. This increases its use as an additive in various food items to improve their properties. After eating a diet rich in maltooligosaccharides, the hydrolysis of MOS to glucose occurs mainly in the small intestine by glucoamylase-maltase [1], which provides a steady source of energy. Therefore, this “maltooligomer mix” (preferably) or pure maltooligosaccharides can be used as a component in nutritional supplements for athletes and some special patients [6].

MOS are widely used in the bakery industry as antistaling agents due to their ability to prevent starch retrogradation. The retrogradation of starch is caused by an increase in its crystallinity due to the progressive transition of the starch from its amorphous structure to the partially crystalline form through the intramolecular or intermolecular association of starch molecules via hydrogen bonding. The retrogradation of starch decreases the freshness of baked goods because the crumb becomes hard, and this phenomenon is called stalling of baked goods [20]. Stopping or decreasing the firming of bread can be prevented by adding amylases, which can produce maltooligosaccharides and maltodextrins in the bread, or by directly adding maltooligosaccharides. For this, maltooligosaccharides with a DP lower than 6 are used because those with a DP higher than 6 can form small helical structures that co-crystallize with starch and increase retrogradation [21]. Many MOS are also used as chiral selectors for separating various substances, e.g., pharmaceuticals, by capillary electrophoresis. In such cases, maltooligosaccharides with a higher DP and maltodextrins are used [22].

### 3.2. Derivatives of Maltooligosaccharides

Derivatization of MOS opens the way to the preparation of a virtually unlimited spectrum of new substances with tailored properties depending on the bound substituent. Thus, by choosing a suitable substituent the application of a MOS derivate might be more specific than the application of maltooligosaccharides in their natural form.

There is a wide range of reactions by which carbohydrates can be modified, e.g., sulfation, carboxy-methylation, acetylation, phosphorylation, selenization and alkylation. To the best of the authors’ knowledge, all of these modifications have been reported for MOS, except for selenization. The following chapter provides an overview of the most widely used derivatives of MOS. 

#### 3.2.1. Maltooligosaccharide Derivatives for Spectrophotometric Determination of Enzyme Activity

The chromogenic or fluorogenic derivatives of maltooligosaccharides are widely used for easily determining enzymatic activity by measuring the absorbance or fluo-rescence due to the release of the chromophore/fluorophore from a substrate by the enzymatic reaction. The mentioned derivatives are also very suitable for automated analytical tools. Such assays are performed wherever the presence of α-amylase needs to be determined, for example, in the human serum, urine, various physiologically active substances, and various food products, among others [23]. The most frequently used for alpha-Amylase assay are BpNPG7 (non-reducing-end blocked p-nitrophenyl malto-heptaoside) [24,25,26,27,28,29] and ethylidene-4-NP-G7 (4-nitrophenyl-4,6-ethylidene-alpha-D-maltoheptaoside) [30,31] which are also parts of commertial kits. Beta-Amylase assay using PNPβ-G3 (p-nitrophenyl-β-D-maltotrioside) [27,32] and BpNPG7 [26] was reported as well.

Compared to chromogenic MOS derivatives, fluorogenic derivatives provide several orders of magnitude higher analytical sensitivity and are also used for the investigation of enzymatic activity. This applies for example for fluorogenic pyridylamino maltooligosaccharides that were used for the assay of different enzymes, like glycogen phosphorylase [33], amylo-1,6-glucosidase,4-α-glucanotransferase [34], and glycogen debranching enzyme [35]. Other frequently used substrate types are various derivatives with the methylumbelliferyl group, such as 4,6-O-benzylidene-β-4-methylumbelliferyl-maltoheptaoside that was applied for the assay of α-amylase [36], 4,6-O-benzylidene-4-methylumbelliferyl-α-maltotriosyl (1-6)-α-maltotrioside used as a substrate of pullulanase [37], and 4,6-O-benzylidene-4-methylumbelliferyl-α-6^3^-α-D-maltotriosyl-maltotrioside applied for the dextrinase assay [38].

#### 3.2.2. Alkylmaltooligosaccharides

Therapeutic agents are often combined with various surfactants that are a part of drug carriers. These surfactants are often irritable to the skin and other tissues, including mucous membranes. Many surfactants also cause the denaturation of proteins, destroying their biological activity. Thus, the demand is increasing for ideal surfactants with some important properties, such as non-toxicity and non-irritability. They should facilitate drug absorption across various membrane barriers without compromising the biological function and structural integrity of the membranes. Examples of such promising surfactants are alkylglycosides [39] and alkypolyglycosides.

Due to their amphiphilic structure and excellent surface activity, along with product safety, mildness, multifunctionality, and competitive prices, alkylpolyglycosides are used extensively and are considered to be the most successful sugar-based surfactants. Alkylpolyglycosides were first synthesized more than 100 years ago by Emil Fischer [40]. They are characterized by the length of the alkyl chain and the number of glucose units attached to the chain (Figure 3) [41]. If the glucose units are linked by α-1,4 linkage, and their numbers vary in the range of 3–10 units, they are called alkylmaltooligosaccharides.

Besides alkylmaltooligosaccharides, alkylmaltosides are also potent surfactants with attractive properties. Alkylmaltoside consists of an alkyl chain bonded to an anomeric carbon of maltose. Together with alkylmaltooligosaccharides containing maltotetraose and maltotriose, alkylmaltosides increase the absorption, biological availability, and activity of the drug [39].

#### 3.2.3. Sulfated Alkylmaltooligosaccharides

In 1986, polyanionic compounds, such as dextran sulfate and poly(methacrylic acid), were suggested as potential drugs for HIV [44]. It was the beginning of the development of related compounds with anti-HIV activity. A year later, synthetic sulfated polysaccharides were found to inhibit HIV infection [45]. In 1988–1989, studies reported that synthesized sulfate of lentinan (1,3-β-D-glucan with two 1,6-β-glucopyranoside branches for every five 1,3-β-glucopyranoside linear linkages) had high anti-HIV activity [46,47]. The main disadvantage of the compounds with the anti-HIV activity reported up to that time was their anticoagulant activity, which is undesirable in antivirals. Therefore, the search for better compounds continued [48]. One year later, the same research team published a study reporting the synthesis of sulfate of curdlan (polysaccharide composed entirely of glucose subunits linked by ß-1,3-glycosidic linkage) with high anti-HIV activity and low anticoagulant activity with low toxicity in animal tests. This compound underwent phase I clinical trials, in which the intravenous administration of curdlan sulfate was found to increase CD4 lymphocytes of HIV-infected patients in a dose-dependent manner with no major side effects [48,49].

Although curdlan sulfate was promising, researchers continued to investigate lower molecular weight sulfated oligosaccharides, as they were expected to have lower cytotoxicity. Katsuraya et al. investigated two main groups of sulfated oligosaccharides, including sulfated dodecyl laminari-oligosaccharides and sulfated al-kylmaltooligosaccharides [48]. Since laminari-oligosaccharides (derivatives of laminarin, a compound composed of glucose subunits linked by ß-1,3-glycosidic bonds, with a small amount of branches composed of ß-1,6-glycosidic linked glucose units) are relatively rare, maltooligosaccharides are preferred. However, due to their low molecular weight, they also have low anti-HIV activity. This problem was circumvented by attaching an alkyl chain to the reducing end of the maltooligosaccharide to form a sulfated alkylmaltooligosaccharide (Figure 4). This increased the anti-HIV activity by tens to hundreds of times compared to the anti-HIV activity of free sulfated maltooligosaccharides [50]. Experimental research has shown that, while sulfated polysaccharides have antiviral activity against several enveloped viruses, such as HIV, cytomegalovirus, influenza A, and dengue virus [51], in the case of sulfated maltooligosaccharides only anti-HIV activity has been demonstrated so far.

The anti-HIV activity is also affected by the number of monosaccharide units, and therefore, this effect was investigated. The study showed that sulfated alkylmaltooli-gosaccharides with maltopentaose, maltohexaose, or maltoheptaose in their structure had a higher anti-HIV activity compared to sulfated alkylmaltooligosaccharides with maltotetraose in their structure, which showed very low anti-HIV activity [48]. The high anti-HIV activity was due to the synergistic effect of the alkyl chain, which can penetrate the lipid bilayer of HIV, and sulfated maltooligosaccharides and their electrostatic interaction with the HIV gp120 glycoprotein found on the surface of the virus, which can inhibit HIV infection. In general, the alkyl chain length varies in the range of C10–C18; however, with an increase in the length of the alkyl chain, the cytotoxicity of these compounds also increases. Thus, maintaining a balance between the alkyl chain length and hydrophilic oligosaccharide moieties is important [50].

#### 3.2.4. Sulfated Maltooligosaccharides

Although sulfated maltooligosaccharides without an alkyl chain do not have therapeutic effects against HIV infection, they are potent antitumor agents due to their ability to inhibit heparanase [50]. Heparanase is a glycosyl hydrolase and cleaves the heparan sulfate (HS) side chain of proteoglycans. HS occurs on cell surfaces and is also a major component of the extracellular matrix [52]. Heparanase has many essential functions, for example, it helps in reorganizing tissues during embryonic development and participates in angiogenesis, wound healing, the proliferation of smooth muscles, and other functions [53,54,55,56]. Heparanase also promotes the metastasis of tumor cells and the formation of secondary tumors [53]. Thus, inhibitors of these enzymes are potential drug targets with antitumor and antiangiogenic properties.

Several groups of inhibitors of these enzymes are known, including putative transition state analogs and monoclonal antibodies, but the main group consists of heparin mimetics, including sulfated maltooligosaccharides (Figure 5). Tressler et al. investigated sulfated maltose and maltooligosaccharides with 3–7 monosaccharide units. Among them, sulfated maltoheptaose and sulfated maltoheptaose containing an acetyl group (NHAc) at the reducing end were found to be the most efficient inhibitor with the same potency as heparin (IC50 = 4.1 µg/mL). The effectiveness of sulfated maltotetraose was slightly lesser than that of heparin (IC50 = 9.0 µg/mL), and sulfated maltose showed no activity even at a concentration higher than 200 µg/mL [56,57].

#### 3.2.5. Other Derivatives of Maltooligosaccharides

In addition to those described above, several other types of MOS derivatives have been identified, but only some of them have been shown to have a significant impact on their practical use. These minor MOS derivatives are listed in Table 2 together with their applications like enzyme substrates and inhibitors, products for the pharmaceutical or cosmetics industry, materials for the preparation of polymers, and others.

## 4. Overview of Different Ways of Producing Maltooligosaccharides

With an increase in the popularity of maltooligosaccharides because of their unique properties and widespread use, the demand for their synthetic preparation has also increased. Maltooligosaccharides and oligosaccharides cannot be efficiently synthesized by chemical synthesis methods because of the stereospecific glycosidic linkages in their structure. The synthesis of stereospecific glycosidic linkages by chemical methods requires many reaction steps to protect reactive hydroxyl groups. Additionally, such a process of synthesis often requires extreme conditions and the use of toxic catalysts, which is undesirable in the context of environmental sustainability. Along with chemical synthesis from glucose monomers, a method based on acid hydrolysis of starch might also be used. However, this method also has several drawbacks, such as the formation of non-specific products, and thus, the hydrolysis process needs to be strictly regulated. Many commercially available microbial amylases have almost completely replaced the classical acid hydrolysis of starch. The enzymatic synthesis facilitates the whole process and provides additional advantages, such as regiospecificity without the need for protection/deprotection steps, stereospecificity, and mild reaction conditions, which benefit the environment [76,77]. In this study, only enzymatic synthesis is discussed, and henceforth, the word “synthesis” means enzymatic synthesis unless otherwise stated.

The composition of the resulting hydrolysate depends on the temperature, hydrolysis conditions, and the origin of the enzyme, while the selection of the enzyme depends on its properties, which include substrate specificity, thermostability, and pH optimum [18].

Based on the enzyme and the conditions used, maltooligosaccharides can be synthesized in two ways: (i) the synthesis of a mixture of maltooligosaccharides or (ii) the synthesis of predominantly one type of maltooligosaccharide with the desired number of monosaccharide units. These maltooligosaccharides can be used directly in food and dairy products for enriching the final products [78] or the individual forms of maltooligosaccharides can be isolated from this mixture using various techniques, which are discussed later. However, isolating one type of MOS from the MOS mixture produced by the non-specific hydrolysis of starch is often more complicated than purifying the desired form of MOS after a specific reaction. In the second approach, one maltooligosaccharide dominates the resulting reaction mixture as the synthesis of other maltooligosaccharides is suppressed. The isolation process, in this case, is often cheaper and faster. The pure forms of maltooligosaccharides have more specific uses than a mixture of maltooligosaccharides, as mentioned above.

The commercial preparation of maltooligosaccharides involves combining different approaches. However, it results in the production of a mixture of maltooligosaccharides, commercially available as Fuji Oligo syrups. This commercial production of maltooligosaccharides includes three main steps: (i) the liquefaction of starch, (ii) its subsequent saccharification, and (iii) the purification of MOS [6,79,80,81]. The liquefaction of starch uses α-amylases, acids, or the action of physical factorsImpurities, such as proteins and fats, are removed at this stage if required. This step is then followed by the key step of the method, which includes the saccharification of starch to form maltooligosaccharides. The saccharification processes can be divided into a single enzyme process consisting of the use of MFAse and a dual enzyme process that combines the action of MFAse and pullulanase [6]. In the dual enzyme system, adding pullulanase to the reaction system might increase the reaction rate besides increasing the yield of maltooligosaccharides. Pullulanase can be added before saccharification along with MFAse, which can lead to the simultaneous action of both enzymes after the saccharification process.

This process was recently investigated, and the best results were obtained with the addition of pullulanase after saccharification. When pullulanase was added before saccharification, the reaction rate did not increase significantly. When the substrate was treated simultaneously with MFAse and pullulanase, the overall conversion rate was the lowest [10].

The last important step in the industrial preparation of maltooligosaccharides involves purification, by which impurities, such as unconsumed starch, maltodextrins, and enzymes, are removed.

Apart from industrial production, in general, the enzymatic synthesis of maltooligosaccharides can be performed by two types of reactions, including hydrolysis and transglycosylation. The following section discusses the preparation of maltooligosaccharides by glycosyl hydrolases and glycosyl transferases. These preparation approaches are also summarized in Table 3, which was created by concatenating and extracting information from papers published recently [6,7,11]. Other examples of enzymes used for producing MOS are also presented in the table and mentioned in the following sections.

### 4.1. Preparation of MOS by Hydrolytic Reactions

Enzymes involved in the conventional method for preparing maltooligosaccharides are glycoside hydrolases which catalyze hydrolytic reactions of glycosidic bonds in complex carbohydrates. In the conventional method, debranching enzymes, such as pullulanase (EC 3.2.1.41) and isoamylase (EC 3.2.1.68), initially act on starch, hydrolyzing the side linkages. This process is followed by hydrolysis using various α-amylases [125]. However, this method has some disadvantages, for example, the reaction conditions need to be controlled precisely for efficiently synthesizing maltooligosaccharides with the desired number of monosaccharide units [77].

In general, two types of hydrolytic enzymes are used for the synthesis of maltooligosaccharides, including α-amylases and maltooligosaccharide-forming amylases.

#### 4.1.1. α-Amylases

The α-amylases (EC 3.2.1.1) belong to glycosyl hydrolase family 13 (GH 13) and catalyze the hydrolysis of α-1,4-glycosidic linkages in starch and related α-1,4-glucans, which leads to the formation of glucose or maltose [126].

Since α-amylase hydrolyzes starch to glucose and maltose as the final products, for preparing maltooligosaccharides, the hydrolysis needs to be controlled, and the reaction needs to be stopped at the stage in which the content of the desired maltooligosaccharides is the highest. The final composition of the reaction mixture after hydrolysis also strongly depends on the hydrolysis conditions and the origin of the enzyme. However, this method of MOS production with a specific DP is inefficient since a mixture of various carbohydrates, predominantly containing MOS, is commonly formed. Such a mixture of MOS is then often used directly to enrich different products in the food industry or can be further processed for isolating individual maltooligosaccharides.

Beyond the conventional method, α-amylases are widely used in other ways to produce maltooligosaccharides. Moon and Cho (1997) produced maltopentaose (G5) from starch by α-amylase (commercially called Termamyl^®^) by modifying the reaction conditions. Although maltopentaose was the most dominant product, the highest selectivity for the production of G5 by Termamyl was only 40%, and therefore, the product needed to be isolated. The isolation of G5 from the mixture of other types of maltooligosaccharides by adsorption on activated carbon yielded a maximum G5 content of 70% [82].

In another way of producing maltooligosaccharides, black potato starch was used, and maltotriose was the main product formed by the action of α-amylase obtained from *Brevibacterium* sp. [83]. However, the authors did not specify the type of amylase used.

Several other cases of the production of maltooligosaccharides from starch using α-amylases from marine bacteria are also known. For example, α-amylase from *Chromohalobacter* sp. TVSP 101 produces mainly glucose, maltose, maltotriose, and maltotetraose [84], α-amylase from the halophilic *Saccharopolyspora* sp. A9 can catalyze the production of glucose, maltose, maltotriose [85], and α-amylase from *Marinobacter* sp. EMB8 yields the highest content of glucose, maltose, maltotriose, and maltotetraose [105].

The properties and effectiveness of maltooligosaccharides strongly depend on their degree of polymerization. Industrially produced maltodextrins, maltooligosaccharides, and starch hydrolysates, when prepared by non-specific α-amylases, commonly consist of a mixture of maltooligosaccharides that have different degrees of polymerization. The product formed is therefore characterized by the properties resulting from the MOS mixture, rather than the properties of a single type of MOS, which might be required for specific applications. Therefore, several studies have investigated ways to produce maltooligosaccharides and maltodextrins with a narrow distribution of the DP [95] and found that it can be achieved by using maltooligosaccharide-forming amylases.

#### 4.1.2. Maltooligosaccharide-Forming Amylases

For enhancing the production of maltooligosaccharides using synthetic strategies, maltooligosaccharide-forming amylases have been identified and implemented. These enzymes include maltotriose-forming amylase (G3-amylase, EC 3.2.1.116), malto-tetraose-forming amylase (G4-amylase, EC 3.2.1.60), maltopentaose-forming amylase (G5-amylase, EC 3.2.1.-), and maltohexaose-forming amylase (G6-amylase, EC 3.2.1.98). Together, they are called maltooligosaccharide-forming amylases (MFAses) and belong to GH13. Since the 1970s, different types of MFAses have been identified in various microorganisms. MFAses can differ in structure, properties, and production specificity [6].

Although the overall structure of MFAses is similar to that of other α-amylases from GH 13, MFAses preferably produce one type of maltooligosaccharide. Their ability to generate maltooligosaccharides from starch or related α-1,4-glucans instead of further cleaving the product to monosaccharides (glucose) or disaccharides (maltose) is also their main difference from other α-amylases (EC 3.2.1.1) [127]. MFAses might be exo-acting amylases that hydrolyze the substrate from the non-reducing end with high specificity for only one product. Besides exo-acting MFAses, there are also maltoooligosaccharide-forming amylases that non-specifically form maltooligosaccharides and are mostly endo-acting amylases that randomly cleave internal α-1,4 linkages [6]. However, amylases that specifically generate one type of maltooligosaccharide are more attractive.

### 4.2. Preparation of MOS by Transglycosylation Reaction

Transglycosylation leads to the formation of glycosidic linkages between the starting units. The product has a higher molecular weight than the initial substrate, and the linkage is formed by transferring a group between the group donor and the group acceptor. Transglycosylation can be performed by glycosyl hydrolases or by glycosyltransferases (Table 3). Although transglycosylation is not the primary reaction catalyzed by glycosyl hydrolases, it can be achieved by changing the reaction conditions, e.g., by reducing the water activity in the presence of a suitable acceptor other than water [128]. However, glycosyltransferases only catalyze transglycosylation, unlike glycosyl hydrolases.

The term transglycosylation was first introduced in 1951 by Hehre [129]. Although amylases are the most widely used enzymes for preparing maltooligosaccharides, transglycosylation reactions might also be used for this purpose. According to some researchers, transglycosylation reactions also have some advantages over conventional approaches [77], such as a higher yield of oligosaccharides with a narrower range of the DP of the final MOS and better regulation of the process. Another significant advantage is the less tedious production of MOS with a higher DP compared to the conventional enzymatic hydrolysis of starch by α-amylase. 

#### 4.2.1. α-Amylases

The ability of the enzyme to catalyze transglycosylation and hydrolytic reactions was first observed with levansucrase [130]; however, the dual activity of the enzyme was demonstrated in 1950 with two well-known hydrolases, i.e., yeast invertase [131,132] and ß-glucosidase [133]. Initial studies on the catalytic mechanisms of transglycosylation of α-amylases were conducted in the early 1970s [134]. Their transglycosylation capability is now well-known, and they are applied in different ways to synthesize maltooligosaccharides.

Kobayashi et al. described the production of MOS by the transglycosylation re-action of α-amylase from *Aspergillus oryzae* using fluorinated α-D-maltosyl at a suitable concentration. The reaction mixture consisted of various water-soluble organic solvents and a buffer solution in an appropriate ratio [77].

Maltogenic amylase (MA) (EC 3.2.1.133) is also a hydrolase, but at low water activity, it can transglycosylate like other hydrolases. Water activity can be reduced by (i) the increase in the substrate concentration or (ii) the addition of an organic solvent to the reaction mixture. However, these conditions are disadvantageous for industrial production because increasing the substrate concentration is uneconomical at an industrial scale, and the use of high amounts of an organic solvent should be avoided, especially when the final product is to be used in the food industry.

Manas et al. performed site-directed mutagenesis and used a rational design to change the architecture of the active site of the enzyme (MFAse), which favored the transglycosylation reaction without requiring any modification of this condition [135].

#### 4.2.2. Cyclodextrin Glycosyltransferases

Cyclodextrin glycosyltransferases (CGTases) (EC 2.4.1.19) are placed in the α-amylase family (GH 13) because of the similarities in their structure domains; however, they possess different catalytic properties from those of α-amylases. CGTase is a bacterial enzyme that can catalyze three transferase reactions, including (i) cyclization, in which the starch is converted into cyclodextrins (CDs), (ii) coupling, which cleaves the CD molecule and couples it with a linear oligosaccharide to form a longer linear oligosaccharide, and (iii) disproportionation, which produces longer oligosaccharides by transferring a part of a linear oligosaccharide to another linear saccharide acting as an acceptor [136].

The CGTase from *Thermoanaerobacter* sp. has the highest recorded disproportionation activity. The ratio of donor to acceptor is critical for alternating the selectivity of the process and reducing cyclodextrin synthesis. This method of synthesizing short-chain MOS uses only one enzyme, which is better than the classical conventional industrial process, where at least two enzymes are usually required [136].

Another example includes the CGTase from *Bacillus circulans* DF 9R, which catalyzes MOS synthesis by intramolecular transglycosylation. The reactor operation in a continuous regime can provide a 12% higher yield than the batch process. Adding glucose or maltose to the reaction mixture can further increase MOS production [123].

### 4.3. Preparation of MOS Combining Hydrolytic and Transglycosylation Reactions

Several other methods are used for MOS production. One such method involves combining the transglycosylation activity of amylosucrase from *Cellulomonas carbonis* (CC-ASase) with the hydrolytic activity of α-amylases from *Pseudomonas mendocina* (PM-Amy); these reactions are synergistic. Amylosucrase is a transferase (EC 2.4.1.4) that can synthesize α-1,4-linked glucans using only sucrose as the substrate. According to the described method, α-1,4-linked glucans synthesized by CC-ASase are further cleaved by the hydrolytic action of PM-Amy to form MOS (Figure 6). These can be performed in a one-pot reaction or two steps, but the productivity is higher in the two-step process, which has a MOS yield of about 34.1% (*w/w*). The ratio of enzyme activities in the one-pot reaction is an important factor. A decrease in the ratio of CC-ASase to PM-Amy activity decreases the production of MOS due to the lack of substrate (α-1,4-linked glucans). Since the method in which starch is used as a substrate requires liquefaction and gelatinization, which are energy-demanding and time-consuming steps, this is an attractive alternative method of MOS production [124].

## 5. Preparation of Maltooligosaccharide Derivatives

### 5.1. Maltooligosaccharide Chromogenic or Fluorogenic Substrates

The first enzymatic process used for producing chromogenic or fluorogenic derivatives of maltooligosaccharides was described by Wallenfels et al. in 1978 [137]. This process uses transferases, for example, cyclodextrin glycosyltransferases, and some chromogenic or fluorogenic derivatives of glucosides as the acceptor (p-nitrophenyl α- and ß-glucosides, 4-methylumbelliferyl α-D-glucoside, and strophantyl α-D-glucoside) and cyclohexaamylose as the donor to form α- or ß-phenyl-, 4-metylumbelliferyl-, and strophantyl-maltooligosaccharides [137].

Another way of preparing such derivatives of maltooligosaccharides was described in 1993 by Usui et al. They used the transglycosylation ability of amylases (for example, glucoamylase, maltotriose-producing amylase, maltotetraose-producing amylase, maltopentaose-producing amylase, maltohexaose-producing amylase, etc.) in the presence of a mixture of maltooligosaccharides and o-glucosyl derivative to produce high purity derivatives of maltooligosaccharides. The reaction was conducted in a mixture of hydrophilic organic solvent and water [23,138] to enhance the transglycosylation activity of amylase.

One major limitation of these derivatives of maltooligosaccharides is their susceptibility to substrate cleavage by α-glucosidase, which is commonly found in body fluids. To circumvent this problem, researchers developed a new method using new types of maltooligosaccharide derivatives with hydroxyl groups at positions 6 and 4 substituted by an alkyl-, alkoyl-, or phenyl group, making them poorly soluble in water, or by methylsulfinylethyl, ethylsulfinylethyl, methanesulfonylethyl, ethanesulfonylethyl, etc., 2-ketopropyl, 2-ketobutyl, 3-ketobutyl, 2-ketopentyl, 3-ketopentyl, 4-ketopentyl, etc., which have better solubility in water. These substituents prevent the compound from being degraded by additional enzymes, and the reaction can thus be performed using a one-part liquid reagent for precisely determining the α-amylase activity [139].

Further modifications of the oligosaccharide moiety, in which the non-reducing end of 4-nitrophenyl or 2-chloro-4-nitrophenyl maltooligosaccharides (DP 4-8) is blocked by ethylidene [140] or benzylidene [141] compounds, have improved the efficiency of chromogenic substrates. Colorimetric assays using these compounds are more stable and are currently used to measure the enzyme activity of human salivary and pancreatic α-amylases (see the Section 3.2.1).

More recently, Oka et al. described a novel synthetic route for construction of bi-fluorescence-labeled maltohexaoside [142] and later extended this method and synthesized of a series of bi-fluorescence labeled MOS (DP 4-7) as potential fluorogenic substrates for the rapid detection of α-amylase activity to study various amylase-related diseases [143].

### 5.2. Alkylmaltooligosaccharides

Rogers and Barresi showed that maltooligosaccharides are excellent starting materials for producing glycosides with a higher degree of polymerization. If glucose is used for glycosylation, the resulting glycosides are limited to an average DP of 1.3, and therefore, glucose is unsuitable for producing glycosides with a higher level of polymerization. In contrast, alkylglycosides with a higher DP cannot be prepared from starting materials with a higher DP, such as starch, because the reaction between starch and alcohol results in the formation of alkylglycosides with a low DP only [144,145].

Rogers and Barresi showed that maltooligosaccharides have a degree of polymerization greater than 10 and some maltooligosaccharides may have a DP as high as 50. However, they stated that MOS with a DP of 5–25 were the most suitable for glycosylation. In that study, the maltooligosaccharides were glycosylated by alcohol (preferably methanol or ethanol) or thiol. In the case of glycosylation with thiol, the thiol nucleophile was preferred. The glycosylation reaction was catalyzed by an acid catalyst like p-toluenesulfonic acid. They found that using maltooligosaccharide mixtures produces a unique mixture of maltooligosaccharide-derived glycosides, unlike starch. When the right mixture of MOS is used as the starting material, with suitable reaction conditions like the reaction time and temperature, the level of an acid catalyst, and the percentage of solid substrate material in the mixture, alkylglycosides with the desired DP might be obtained [144].

## 6. Downstream Processing of Maltooligosaccharides

The isolation of maltooligosaccharides after their synthesis is the next challenge in their preparation.

Several purification techniques are used for obtaining pure forms of maltooligosaccharides from enzymatically synthesized mixtures. The suitability of a purification method is governed by (i) the composition of the mixture before purification, (ii) the final application of the purified compound, (iii) the economic feasibility, and (iv) the available equipment.

Mixtures of maltooligosaccharides can be fractionated by gel permeation chromatography. Two main types of gels can be used for separating maltooligosaccharides from the mixture, including polyacrylamide gel and hydrophilic vinyl polymer gel. The polyacrylamide gel was first used for separating maltooligosaccharides in 1969 [146]. Since then, several researchers have investigated the influence of temperature on the distribution coefficient [147] or the elution order of individual molecules [148]. Pure maltooligosaccharides cannot be obtained in large volumes by this method, especially maltooligosaccharides with a DP ≥ 7. Thus, other methods have been developed.

The hydrophilic vinyl polymer gel is an alternative method developed in the 1990s for separating maltooligosaccharides. Several researchers have published studies on the influence of various factors on separation, such as flow rate and bed height. However, this method was also found to be unsuitable for the separation of MOS with a DP > 6 [149]. Additionally, during the preparation of MOS (DP 6–8) by the partial hydrolysis of cyclodextrins, the MOS resolution and column loading capacity were found to be low and, thus, unsuitable for industrial use [150]. Methods for purifying MOS by gel permeation chromatography were not further developed because more efficient preparative fractionation methods are available.

The first technique of preparative fractionation of maltooligosaccharides involved their selective adsorption on activated carbon and celite. In that method, a mixture of charcoal and celite was used as the stationary phase, and water/ethanol was used as the mobile phase to eliminate mono-, di-, and trisaccharides from the mixture of maltooligosaccharides [151]. Other techniques were optimized and developed based on this method. Later, maltooligosaccharides and maltopolysaccharides with a DP of up to 18 were purified using cellulose as the stationary phase and butanol as the mobile phase [152]. However, butanol cannot be used in the food industry due to its toxicity, and food-grade butanol is expensive. Therefore, other techniques were developed that did not require butanol as a mobile phase. This resulted in the development of a method suitable for the food industry using a cheap and food-safe mobile phase composed of water and ethanol. Although this method narrowed the wide range of DP, it did not yield pure maltooligosaccharides [153]. Thus, it was later modified, and a successful method was developed only in 2018. Pure maltooligosaccharides were obtained by a relatively simple method based on preparative chromatography with microcrystalline cellulose and different solubilities of maltooligosaccharides and maltopolysaccharides in ethanolic solutions [154]. In this method, relatively simple and inexpensive materials are used, which are also suitable for the food industry (food-grade cellulose, ethanol, and maltodextrin-based products). Thus, this method can be used for the large-scale production of MOS.

Another method that can increase the purity of maltooligosaccharides is selective fermentation. It is an additional step requiring biomass removal and further purification. However, it eliminates contaminating sugars, such as glucose, maltose, and maltotriose. For removing glucose, for total concentrations up to 500 g·L^−1^, the immobilized cells of *Zymomonas mobilis* can be used. *Zymomonas mobilis* can only ferment some carbon substrates, such as sucrose, glucose, and fructose, to produce ethanol and carbon dioxide. Ethanol can be easily separated while drying the final product [155]. To remove saccharides, such as glucose, maltose, and (partially) maltotriose, the yeast *Saccharomyces cerevisiae* can be used [156]. Since higher saccharides remain intact, such an approach might benefit the preparation of maltotetraose, where glucose, maltose, and maltotriose are by-products, and their removal is crucial. However, this strategy might also be used while preparing maltotriose with the directed fermentation of lower saccharides, where the fermentation must be stopped at the right stage to avoid maltotriose fermentation by the yeast.

Membrane technology is another method of purifying oligosaccharides. This method has been used for purifying galactooligosaccharides, fructooligosaccharides, xylooligosaccharides, and isomaltooligosaccharides, but its application in MOS purification remains undescribed. Membrane technology uses the principle of membrane nanofiltration. Saccharides with a higher DP are trapped in the retentate, while monosaccharides and other impurities pass through the membrane into the permeate stream [157]. This is a simple, scale-up, and energy-efficient technology, which can directly immobilize the biocatalyst onto the membrane. However, it has certain limitations, such as high price, sealing problems, mechanical and thermal brittleness of some membranes, and limited experience with this type of separation technique. However, the immobilization of the biocatalyst directly onto the membrane ensures continuous conditions of MOS production with subsequent separation, which is a significant advantage in combining mass transfer and enzymatic reaction. This synthesis design increases the conversion and efficiency of the whole process. Additionally, ap-propriate immobilization of the biocatalyst increases its stability against various stress factors and allows for its continuous reuse, which is a great strategy for process intensification for the large-scale production of oligosaccharides, including MOS [157].

## 7. Immobilization of Enzymes for the Production of Maltooligosaccharides

The implementation of immobilized enzymes into biocatalysis has many advantages, especially concerning green and sustainable production. An appropriate immobilization technique improves the stability and reusability of the enzyme, which is the main benefit of using this method. There are three main types of immobilization: (i) binding to a carrier (support), where the binding can be physical, ionic, or covalent, (ii) entrapment/encapsulation in a carrier, and (iii) cross-linking (carrier-free) immobilization methods. Common carriers used for enzyme immobilization are synthetic resins, biopolymers, inorganic solids, and polyacrylamide, silica sol-gel. Enzymes might also be immobilized in hollow fibers and used in membrane reactors [158].

In this chapter, we reviewed the application of immobilized enzymes in the pro-duction of MOS. These enzymes were previously described in Section 4. Although cyclodextrin glycosyltransferases (CGTases) can be used for synthesizing MOS instead of synthesizing cyclodextrins, they are not discussed in this chapter because there is no record of their immobilization for producing maltooligosaccharides. On the contrary, the authors considered MOS synthesis as a side reaction that they tried to avoid, such as in the case of the immobilization of CGTase from *Bacillus macerans* on polysulfone capillary membrane by covalent bonding. The authors of that article found that when the immobilized CGTase was in large amounts and the residence time of the substrate with the biocatalyst was long, larger quantities of MOS were produced because of the CGTase-catalyzed coupling reaction [159]. Further details on the immobilization of these enzymes are presented in the reviewed studies, where the methods of immobilization of CGTase are well-summarized [160,161].

Many studies have investigated the immobilization of α-amylases. Table 4 presents an overview of the information on immobilized α-amylases that are involved in the hydrolysis of starch and starch substrates, which in turn facilitate the production of MOS.

Although the most commonly used method for amylase immobilization is covalent attachment (Table 4), the immobilization of α-amylases (and enzymes in general) by cross-linking (CLEAs) has become popular. One reason might be the growing interest in nanoimmobilization, where CLEAs might be a useful method. Further information on the nanoimmobilization of α-amylases is provided in a recently published review article [183].

Regarding the immobilization of maltooligosaccharide-forming amylases, despite the growing interest in their biocatalytic reaction, only very few cases of immobilization are known. Interestingly, in those studies, the enzymes were immobilized by physical adsorption and encapsulation, rather than by the commonly used covalent bonding method. The immobilization of enzymes in a porous carrier structure like gels or porous particles leads to mass transfer limitations, which can be highly significant, especially with high-molecular-weight substrates like starch. Mass transfer limitations decrease the efficiency of the enzyme reaction and depend on many factors, such as the size of the substrate molecule, particle pore size and form, size of the immobilized enzyme particles, the activity of the immobilized enzyme, substrate diffusivity, etc. For amylases, this aspect must be considered when designing the immobilized biocatalyst.

Interest in the immobilization of these enzymes is expected to increase in the future with an increase in the demand for MOS and functional foods in the market.

## 8. Conclusions and Prospects

Oligosaccharides, polysaccharides, and glycosides derived from them form highly diverse structures, which have different properties depending on various factors, such as the length of the carbohydrate chain or the type of aglycone. They have important functions in organisms and glycobiology, as they participate in various tasks, such as cell surface interaction with proteins, cell proliferation, and interaction with other glycans, which alter the mechanical properties of plants and microbial cell walls [184]. They have various features and functions, which require further investigation.

Many studies have shown that maltooligosaccharides and their derivatives have great potential, which is mostly unexplored. Therefore, the number of methods to synthesize maltooligosaccharides is increasing; however, their application on a large scale is challenging.

Improving the properties of MFAses by recombinant technologies was one of the key strategies used in the last decade. Microorganisms, especially bacteria, are mainly used to produce MFAses because they are the most accessible for genetic engineering. Additionally, the production of MFAses by bacteria is faster and cheaper compared to that using other microorganisms.

The individual strains can be modified to optimize the production of MFAses, for example, to increase the yield of MFAses. This strategy not only provides better product yields, but also affects the reaction environment and conditions by targeted modification of the genes encoding MFAses to increase their stability at a broader range of temperatures and other conditions [185]. This might have a positive economic and environmental impact [135]. Overall, the application of recombinant technology in the preparation of MFAses and the production of maltooligosaccharides is expected to grow. However, wild-type producers are still required to obtain genes that encode MFAses with unique properties. Such genes need to be investigated from a structural perspective to identify the localizations responsible for the extraordinary characteristics. Subsequently, such studies need to be combined with bioinformatics approaches to create strains with exceptional properties.

## Figures and Tables

**Figure 1 molecules-28-03281-f001:**
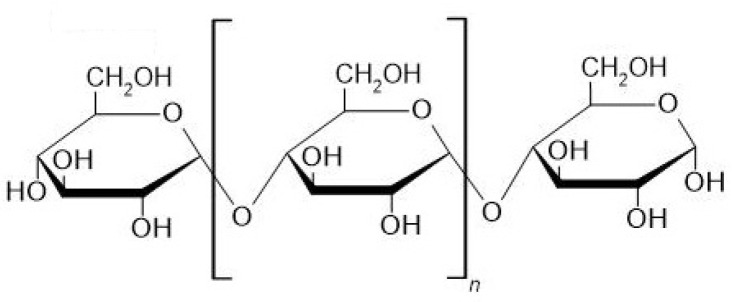
The general structure of maltooligosaccharides.

**Figure 2 molecules-28-03281-f002:**
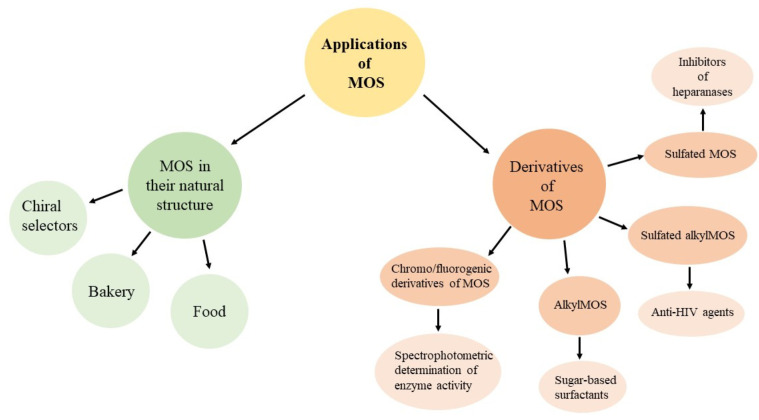
Applications of maltooligosaccharides.

**Figure 3 molecules-28-03281-f003:**
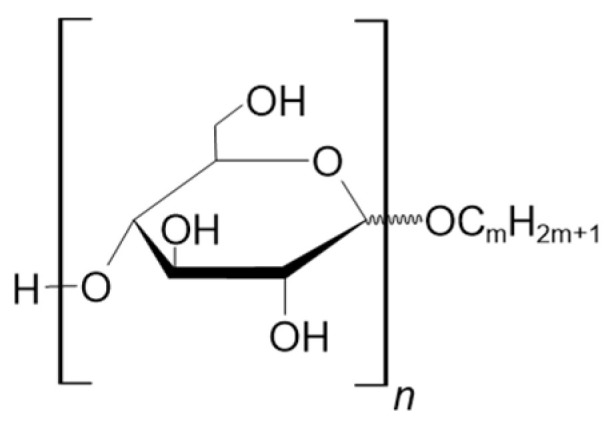
The general structure of alkylpolyglycosides. The index n denotes the number of glucose monomers and varies from 3 to 10. The index m denotes number of carbons in alkyl chain and usually varies from 4 to 18 [42,43].

**Figure 4 molecules-28-03281-f004:**
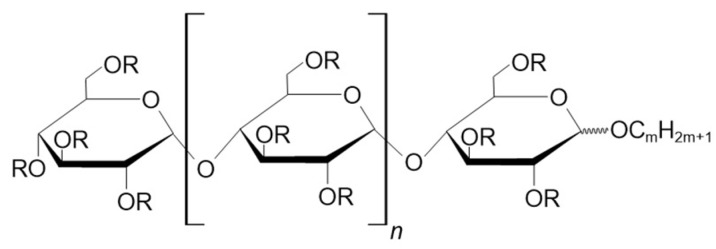
The general structure of sulfated alkylmaltooligosaccharides. R is SO_3_Na or H. The number n varies from 2 to 5 and the number m varies from 12 to 18 [48].

**Figure 5 molecules-28-03281-f005:**
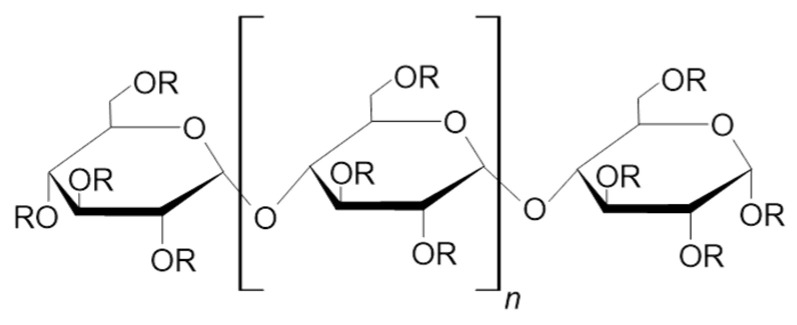
The general structure of sulfated maltooligosaccharides. R is SO_3_Na or H. The number n varies from 2 to 5 [58,59].

**Figure 6 molecules-28-03281-f006:**
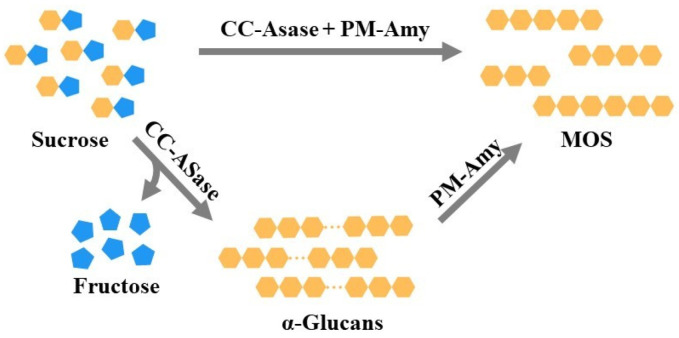
The reaction mechanism associated with MOS production from sucrose. Adapted with permission from Ref. [124]. 2018, Elsevier.

**Table 1 molecules-28-03281-t001:** Basic properties and characteristics of the most common maltooligosaccharides [16,17].

	Maltotriose	Maltotetraose	Maltopentaose	Maltohexaose	Maltoheptaose	Maltooctaose
Number of-glucose unit	3	4	5	6	7	8
CAS Registry Number	1109-28-0	34612-38-9	34620-76-3	34620-77-4	34620-78-5	6156-84-9
Molecular Weight (g·mol^−1^)	504.4	666.6	828.7	990.9	1153.0	1315.1
Water Solubility (g·L^−1^)	554	350	228	252	272	307
Hydrogen Acceptor Count	16	21	26	31	36	41
Hydrogen Donor Count	11	14	17	20	23	26
Refractivity (m^3^·mol⁻^1^)	100.75	133.16	165.58	197.99	230.40	262.82
Polarizability (Å^3^)	46.65	62.12	77.13	92.08	107.11	122.77

**Table 2 molecules-28-03281-t002:** Minor derivatives of MOS.

MOS Derivative	Position of Functional Group	Application	Ref.
Acarviosyl maltooligosaccharides	Non-reducing end	Inhibitors of glycoside hydrolases	[60]
Peracetylated maltooligosaccharides	Both non-reducing and reducing end	Important function as research compound Building block for further synthesis of 4-NP and 2-Cl-4NP-ß-glycosides	[61]
Benzyl maltotriosides	Reducing end	Smooth muscle cell proliferation inhibitor	[62]
Pyridylaminated maltooligosaccharides	Non-reducing end	Activity assay of glycogen phosphorylase and thus study of glycogen phosphorylase mechanism	[33,63]
Carboxylate-terminated maltooligosaccharides	Both non-reducing and reducing end	Cross-linkers of water-soluble chitin to form hydrogel	[64]
3-Azi-1-methoxybutyl-D-maltooligosaccharides	Reducing end	Inhibition of maltose uptake via the maltose-binding protein-dependent transport system in *Escherichia coli*	[65]
α-glucuronylated maltooligosaccharides	Non-reducing end	Represent anionic oligosaccharides useful in glycomaterials	[66]
N-formyl-α-D-glucosaminylated maltopentaoside	Non-reducing end	Glycoscience, potential as drug candidate	[67]
2-amino-2-deoxy-α-D-glucopyranosylated maltooligosaccharides	Non-reducing end	Glycoscience	[68]
Caproyl maltooligosaccharides	6^II^ position: 6th carbon of second glucose unit from reducing end	Biodegradable detergents, fine chemicals in cosmetics, and pharmaceutical industry	[69]
Galactosyl maltooligosaccharidonolactone	Both non-reducing and reducing end	Substrate analogue inhibitors of mammalian α-amylase	[70]
Quercetin-maltooligosaccharides	Reducing end	Food additive or cosmetic ingredient	[71]
2-deoxygluco-maltooligosaccharides	Reducing end	Tracing of the intestinal location of starch	[72]
Carboxymethyl derivatives of p-nitrophenyl-α-maltopentaoside	Both non-reducing and reducing end	Activity assay of α-amylases coupled with glucoamylase and α-D-glucosidase	[73]
Phosphorylated maltooligosaccharides	Non-reducing end-	Monitoring of transport mechanisms in *E. faecalis*; in cosmetics as agents for reduction of appearance and visibility of skin pores	[74,75]

**Table 3 molecules-28-03281-t003:** Enzymes for the production of MOS.

Enzyme Used	Microbial Source	Type of Reaction	Substrate	Optimal Conditions (Temp. °C, pH)	Predominant Product	Ref.
α-Amylase (EC 3.2.1.1) (Termamyl^®^)	-	Hydrolysis	Starch	40, 5.0	G5	[82]
α-Amylase (EC 3.2.1.1)	*Brevibacterium* sp.	Hydrolysis	Starch	-	G1–G3	[83]
	*Chromohalobacter* sp. TVSP 101			37, 9.0	G1–G4	[84]
	*Saccharophyla* sp. A9			-	G1–G3	[85]
	*Bacillus megaterium* VUMB109			93, (-)	G5/G3	[86]
	*Geobacillus stearothermophilus* L07			70, 6.0	G6/G5	[87]
	*Bacillus mojavensis* A21			80, 6.5	G3/G5/G6	[88]
	*Cryptococcus* sp. S-2			50, 6.0	G1–G4	[89]
	*Scytalidium thermophilum* 15.1			60, 6.0	G3–G5	[90]
	*Aspergillus oryzae*	Transglycosylation	Fluorinated α-D-maltosyl	-	G2–G4	[77]
MFAse (EC 3.2.1.-)	*Bacillus koreensis* HL12	Hydrolysis	Starch	40, 7.0	G2–G4	[7]
	*Brachybacterium* sp. LB25			35, 6–7.5	G3	[91]
	*Bacillus circulans* GRS313			48, 4.9	G3/G5	[92]
	*Bacillus subtilis* KCC103			65–70, 6–7	G1–G7	[93]
	*Bacillus subtilis* SDP1			37, 7.0	G2/G3/G5	[94]
	*Bacillus subtilis* US116			65, 6.0	G5/G6/G7	[95]
G3-Amy (EC 3.2.1.116)	*Streptomyces avermitilis* NBRC 14893	Hydrolysis	Starch	-, 6.5	G3	[96]
	*Thermobifida fusca* NTU22			60, 7.0		[97]
	*Fusicoccum sp.* BCC4124			70, 7.0		[98]
	*Microbulbifer thermotolerans* DAU221			50, 6.0		[99]
	*Kitasatospora sp.* MK-1785			55, 6.5		[100]
	*Bacillus subtilis* G3			50, 6–7		[101]
	*Natronococcus sp.* Ah-36			55, 8.7		[102]
G4-Amy (EC 3.2.1.60)	*Bacillus halodurans* MS 2-5	Hydrolysis	Starch	60–65, 10.5–11.0	G4	[103]
	*Pseudomonas stutzeri* AS22			60, 8.0		[104]
	*Marinobacter* sp. EMB8			45, 7.0	G3/G4	[105]
	*Bacillus* sp. GM8901			60, 11–12	G4	[106]
	*Chloroflexus aurantiacus* J-10-F1			71, 7.5	G3/G4	[107]
	*Pseudomonas saccharophila* IAM1504			-	G4	[108]
	*Pseudomonas* sp. IMD 353			50, 7.0		[109]
	*Pseudomonas* sp. MS300			40, 6.8–8.9		[110]
	*Pseudomonas stutzeri* MO-19			50, 7.0		[111]
	*Pseudomonas saccharophila* STB07			-		[112]
G5-Amy (EC 3.2.1.x)	*Bacillus* sp. JAMB-204	Hydrolysis	Starch	60, 6.5	G5	[113]
	Pseudomonas sp. KO-8940			-	G5	[114]
	*Bacillus licheniformis* NCIB 6346			70–90, 7.0	G5	[115]
	*Bacillus stearothermophilus*			60, 5.5	G5/G6	[116]
G6-Amy (EC 3.2.1.98)	*Corallococcus* sp. EGB	Hydrolysis	Starch	50, 7.0	G6	[4]
	*Aerobacter aerogenes*			45, 7.0	G6	[117]
	*Bacillus halodurans* LBK34			60, 10.5–11.5	G6	[118]
	*Bacillus* sp. 707			45, 8.8	G6	[119]
	*Bacillus* sp. H-167			-	G6	[120]
	*Klebsiella pneumonia* IFO-3321			-	G6	[121]
	*Bacillus stearothermophilus* US100			82, 5.6	G6	[122]
CGTase (EC 2.4.1.19)	*Bacillus circulans* DF 9R	Transglycosylation	Glucose/maltose	-	≤G10	[123]
Amylosucrase (EC 2.4.1.4) and α-amylase (EC 3.2.1.1)	*Cellulomonas carboniz* and *Pseudomonas mendocina*	Transglycosylation and hydrolysis	Sucrose	40, 7.0 and 55, 9.0	G3/G4	[124]

**Table 4 molecules-28-03281-t004:** Summary of selected immobilized α-amylases and MFAses. Starch was used as a substrate in all studies with one exception, reduced shortchain amylose in the study of immobilized G4-Amy on Diaion HP-50 [162].

Immobilized Biocatalyst	Immobilization Technique	Source of Biocatalyst	Support	Optimal Conditions (Temp. °C, pH)	Activity of Immobilized Enzyme (U/g Support)	Reusability Studies (% of Initial Activity/Cycles)	**Product Distribution**	**Ref.**
α-Amylase	Covalent attachment	*Aspergillus oryzae*	Corn grits	-	502	-	G1 (8.5%)	[163]
							G2 (66.9%)	
							G3 (18.8%)	
							G4 (2.0%)	
			Porous silica ^(a)^	-	950	-	G1 (20.6%)	
							G2 (68.5%)	
							G3 (3.4%)	
							G4 (4.8%)	
		*Bacillus alcalophilus*	Iron-oxid Magnetic Nanoparticles	-, 8	-	71%/10	-	[164]
		*Bacillus amyloliquefaciens*	Polyaniline Silver Nanoparticles	60, 6.0	-	>80%/10	-	[165]
		*Bacillus licheniformis*	Poly(HEMA-GMA-1-3) membranes	60, 6.5	390	-	-	[166]
			PVA-FEP (PVA-coated poly(tetrafluoroethylenehexafluoropropoylene), CDI method ^(b)^	-, 8.4	525	97%/4	-	[167]
			Sepharose 4B, CNBr method	-, 8.5	1748	85%/4	-	
		*Bacillus* sp.	Zirconium dynamic membrane, GA method	41, 5.5	59.8	-	G2, G3, G4	[168]
		*Cryptococcus flavus*	Glass tube	50, 4.5	-	47%/10	-	[169]
		*Saccharomyces cerevisiae*	Calix[4]arene	60, 7.0	-	62%/10	-	[170]
		Malt	Chitosan-Fe_3_O_4_ CSM	35, 7.0	-	50%/10	-	[171]
	Ionic Attachment		Chitosan-ZnO CSZ	35, 6.0	-	60%/10	-	
		*Bacillus amyloliquifaciens* TSWK1-1	DEAE cellulose	60, 5.5	2186	96%/20	-	[172]
	Covalent attachment		Gelatin	60, 5.5	1771	83%/20	-	
	Entrapment		Polyacrylamide	60, 5.5	1563	65%/20	-	
			Agar	60, 5.5	1600	71%/20	-	
		*Bacillus circulans* GRS 313	Calcium alginate beads	57, 4.9	25.6	85%/7	-	[173]
	Adsorption	*Bacillus subtilis*	Alumina powder	-, 6.0	-	-	-	[174]
	**CLEAs**/crosslinking agent							
	-	*Aspergillus fumigatus*	CL GA–1.5% ^(c)^ (*v/v*)	60, 7.0	-	13%/10	-	[175]
		*Bacillus licheniformis*	CL/BSA + GA	95, 5.5	-	76%/10	-	[176]
		*Bacillus subtilis*	CL/Starch	55, 5.5–7.0	-	>75%/10	-	[177]
			CL/BSA	50, 4.5–7.0	-	>80%/10	-	
			CL GA/0.2% ^(c)^ (*v/v*)	55, 5.5	-	>70%/10	-	
		*Bacillus* sp.	CL GA–0.37% ^(c)^ (*v/v*)	50, 6.0	-	25%/6	-	[178]
			CL on magnetic nanoparticles GA–0.37% ^(c)^ (*v/v*)	60, 6.0	-	100%/6	-	
		-	CL/Agar	-	-	70%/5	-	[179]
			CL/Chitosan	-	-	74%/5	-	
			C L/Dextran	-	-	79%/5	-	
			CL/Gum Arabic	-	-	68%/5	-	
			CL/GA	-	-	63%/5	G4	
G4-Amylase + pullulanase	Physical adsorption	G4-Amylase (G4A): *Pseudomonas stutzeri* NRRL B3389 mutant, pullulanase (P): *Klebsiella pneumoniae*	Chitosan Beads–Chitopearl BCW Series	60 (G4A)/55 (P), 7.0 (G4A)/6.0 (P)	307 (G4A)/84.9 (P)	-	G4	[180,181]
			Duolite S-762	-	271 (G4A)/74.9 (P)	-		
			Diaion HP-50	-	221 (G4A)/44.2 (P)	-		
G4-Amylase		*Pseudomonas stutzeri* NRRL B3389 mutant		55, -	251	-	G4 (45%, in later stages of reaction also G1-G3)	[162]
G6-amylase	Encapsulation by radiocopolymerization	*Aerobacter aerogenes* UV-mutant	polymer of acrylamide + NN’-methylene bis acrylamide (AA+Bis), calcium acrylate (ACa), sodium acrylate (ANa)	53, 7.0	-	100%/20	G6 (in later stages of reaction also G1-G5	[182]

^a^ Various porous silica were tested, whereas here silica with specific area 180 m^2^/g is specified, which has shown the best results among porous silica supports; ^b^ four immobilization methods were tested for the PVA-FEP support, among them only CDI is stated because of the detailed study; ^c^ final volumetric concentration of glutaraldehyde in the solution.

## Data Availability

The data presented in this study are all from published papers and can be obtained from references.

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
