# Peer review of "Maltooligosaccharides: Properties, Production and Applications"

_molecules, 2023, doi:10.3390/molecules28073281_

Round 1

Reviewer 1 Report

The review gives a deep insight into maltooligosaccharides – their applications and production. It is a long review, but I enjoyed reading it, and I found it very well written. It both covers the past and the future. I think researchers with different backgrounds and industry can benefit from the comprehensive review.

I have the following minor comments and suggestions:

Table 1: I would recommend that you just refer to the actual number of glucose units, rather than “Number of n (Figure 1)”. Furthermore, it might be unnecessary to include the chemical formula.

Figure 2: Please define “m” – maybe give a range.

On page 5, you mention different polysaccharides/oligosaccharides (e.g. lentinan, curdlan, and laminari-oligosaccharide), I think that you need to describe those in a few words, so the readers know their composition.

Maybe consider if a few more model structures of the different derivatives, could help the reader. They could be included in figure 3.

Lines 363-366. Here you introduce Table 2. However, I think that you need to introduce it earlier in the text, as the table is not only covering the TG reactions, but also (to a greater extent) the enzymes conducting the relevant hydrolysis reactions.

Reviewer 2 Report

The review "Maltooligosaccharides: properties, production and applications" by the authors M. Bláhová, V. Štefuca, H. Hronská, M. Rosenberg is devoted to the actual problem of systematization and generalization of scientific information about more and more demanded maltooligosaccharides and their derivatives.However, the manuscript needs a revision.

So, in the Introduction, it is necessary to briefly report on the latest analytical reviews already available in this area, as well as to indicate the original role and place of the proposed present review among those already available.

Section 3.1. is literally called “The natural structure of maltooligosaccharides”, which automatically orients the discussion in this section of the features of the natural structure of maltooligosaccharides. However, in this section, there is practically no discussion of any natural structure of maltooligosaccharides, but rather the various uses of natural maltooligosaccharides. Therefore, the title of this thematic section is better reformulated, for example, “The maltooligosaccharides with natural structure”.

Section 3.2.1. “Maltooligosaccharide derivatives for spectrophotometric determination of enzyme activity” is devoted to a very large and popular area of practical application of maltooligosaccharides for colorimetric and fluorrimetric determination of enzyme activity. However, in this section, when discussing many different aspects - from chromogenic or fluorogenic derivatives of maltooligosaccharides, to the determination of alpha-amylase in human blood serum, in urine, in physiologically active substances, in foods, etc., only one single reference [21] is given. This is a fairly old book from 1988. It should be noted that the purpose of the reviews is an analysis of modern information reflected in original works. Therefore, this section of the review should be cardinally supported by an analysis of modern original articles both on the colorimetric determination of enzyme activity and on the fluorescent determination of this activity.

The same applies to section 5.1. “Maltooligosaccharide chromogenic or fluorogenic substrates”, where only three literary references with datings of the last century are given. This section also needs to be fundamentally supported by an analysis of contemporary original articles.

It is surprising, why the authors, without any explanation, immediately operate with only sulfated maltooligosaccharides and alkylmaltooligosaccharides, while esters of other mineral acids, for example, phosphates, etc., are not mentioned at all. If the esters of other mineral acids are not yet known at all for maltooligosaccharides, in contrast to other oligo- and polysaccharides, then this should be emphasized all the more in order to clarify the analysis of only sulphate esters of maltooligosaccharides.

One section 3.2.3 “Sulfated alkylmaltooligosaccharides” is devoted to the biological activity of sulfated alkylmaltooligosaccharides, where only anti-HIV activity is discussed. Against the background of the well-known diverse antiviral activity of other sulfated polysaccharides, the question arises: does the antiviral activity of sulfated alkylmaltooligosaccharides not manifest itself against viruses other than HIV? And if this is true, then this fact should be emphasized all the more, justifying the consideration of antiviral activity solely on the example of HIV.

The authors need to pay attention to the design of Table 3, in particular, to the captions under the table. Here, after the inscriptions of paragraphs a), b), you must either put a semicolon instead of a period, or the inscription of each new paragraph must begin with a capital letter.

Round 2

Reviewer 2 Report

I believe that the revised work, taking into account the previous proposals, can be published.